# Influenza Vaccination and COVID-19 Outcomes in People Older than 50 Years: Data from the Observational Longitudinal SHARE Study

**DOI:** 10.3390/vaccines10060899

**Published:** 2022-06-04

**Authors:** Nicola Veronese, Lee Smith, Francesco Di Gennaro, Olivier Bruyère, Lin Yang, Jacopo Demurtas, Stefania Maggi, Shaun Sabico, Nasser M. Al-Daghri, Mario Barbagallo, Ligia J. Dominguez, Ai Koyanagi

**Affiliations:** 1Geriatric Unit, Department of Internal Medicine and Geriatrics, University of Palermo, 90133 Palermo, Italy; mario.barbagallo@unipa.it (M.B.); ligia.dominguez@unipa.it (L.J.D.); 2Center for Health, Performance and Wellbeing, Anglia Ruskin University, Cambridge CB1 1PTT, UK; lee.smith@anglia.ac.uk; 3Clinic of Infectious Diseases, University of Bari, 70121 Bari, Italy; francesco.digennaro1@uniba.it; 4Department of Public Health, Epidemiology and Health Economics, University of Liège, 4000 Liège, Belgium; olivier.bruyere@uliege.be; 5Department of Cancer Epidemiology and Prevention Research, Cancer Care Alberta, Alberta Health Services, Edmonton, AB T5J 3E4, Canada; lin.yang@albertahealthservices.ca; 6Departments of Oncology and Community Health Sciences, Cumming School of Medicine, University of Calgary, Calgary, AB T2N 1N4, Canada; 7Primary Care Department, Azienda USL Toscana Sud Est, 52100 Grosseto, Italy; jacopo.demurtas@unimore.it; 8Clinical and Experimental Medicine PhD Program, University of Modena and Reggio Emilia, 41121 Modena, Italy; 9Institute of Neuroscience, National Research Council, 00185 Padova, Italy; stefania.maggi@in.cnr.it; 10Chair for Biomarkers of Chronic Diseases, Biochemistry Department, College of Science, King Saud University, Riyadh 11451, Saudi Arabia; ssabico@ksu.edu.sa (S.S.); ndaghri@ksu.edu.sa (N.M.A.-D.); 11Faculty of Medicine, University of Kore, 94100 Enna, Italy; 12Parc Sanitari Sant Joan de Déu/CIBERSAM, ISCIII, Universitat de Barcelona, Fundació Sant Joan de Déu, Sant Boi de Llobregat, 08014 Barcelona, Spain; ai.koyanagi@sjd.es; 13ICREA, 08010 Barcelona, Spain

**Keywords:** influenza, vaccination, COVID-19, SHARE study

## Abstract

Existing literature on the association between influenza vaccination and COVID-19 infection/outcomes is conflicting. Therefore, we aimed to investigate the association between influenza vaccination and COVID-19 outcomes in a large cohort of adults who participated in the SHARE (Survey of Health, Ageing, and Retirement in Europe). Information regarding influenza vaccination in the previous year, and medical and demographic characteristics, were self-reported. Positivity for COVID-19, symptomatology, and hospitalization were also ascertained using self-reported information. An adjusted logistic regression analysis (including 15 baseline factors or propensity score) was used to assess the association between influenza vaccination and COVID-19 outcomes. A total of 48,408 participants (mean age 67 years; 54.1% females) were included. The prevalence of influenza vaccination was 38.3%. After adjusting for 15 potential confounders, influenza vaccination was significantly associated with a lower risk of positivity for COVID-19 (OR = 0.95; *p <* 0.0001), symptomatic forms (OR = 0.87; *p <* 0.0001), and hospitalization for COVID-19 (OR = 0.95; *p <* 0.0001). The results were similar when using a propensity score approach. In conclusion, influenza vaccination may be beneficial for the prevention of COVID-19, as the present study found that influenza vaccination was associated with a small/moderate lower risk of COVID-19 infection and adverse outcomes.

## 1. Introduction

During the last two years, approximately 500 million people were affected by the coronavirus-19 disease (COVID-19), resulting in approximately six million deaths, making this condition a public health priority [1]. The available epidemiological data indicate that COVID-19 could be considered as an emerging geriatric condition [2], because its prevalence and mortality are higher in old compared to young adults [3]. Vaccination against COVID-19 has been reported to be effective in reducing the incidence and severity of COVID-19, also taking into account the variants of the SARS-CoV-2 virus [4].

Moreover, COVID-19 can present with symptoms similar to influenza, one of the most common seasonal viral infections [5,6]. The co-existence of these two conditions may result in a more severe clinical course, and a higher risk of complications or fatal outcomes, as compared to having either one of the conditions [5,7]. Furthermore, people who are particularly vulnerable to COVID-19 and seasonal influenza in terms of unfavorable prognosis share similar characteristics: frail individuals such as older persons and persons with co-morbidities [8,9].

The role of influenza vaccination has been extensively discussed during the COVID-19 epidemic. From one side, influenza vaccination is considered to be of critical importance for high-risk groups not only for reducing influenza complications, but also those of co-infection with the SARS-CoV-2 virus [8]. At the same time, several studies have reported a possible unfavorable relationship between influenza vaccination and coronavirus infections, mainly based on the phenomenon of vaccine-associated virus interference. Recent literature has reported that vaccinated individuals may be at increased risk for other respiratory viruses since they did not receive the non-specific immunity associated with natural infection [10]. For example, influenza vaccination may reduce the probability of contracting influenza, thereby avoiding the activation of the innate immune response [11]. From an epidemiological point of view, however, one ecological study conducted in Italy before the vaccination campaign against SARS-CoV-2 reported that influenza vaccination was associated with a lower risk of all-cause mortality in older people [12]. Another cross-sectional study of approximately 3000 participants, also from Italy, reported that influenza vaccinations did not appear to be associated with COVID-19 prevalence [11].

Although these studies advanced our knowledge regarding the importance of influenza vaccination on COVID-19, they suffer from some limitations such as being conducted in a single country (i.e., Italy) with a small sample size and the lack of individual-level data in an ecological study. Moreover, the possible effect of the vaccination against COVID-19 was not analyzed since this was still not available at the time of these previous studies. More recently, some systematic reviews reported that influenza vaccination was not associated with any increased risk of COVID-19 infection [13] or with a reduced incidence of COVID-19, but not to a decreased risk in more severe forms [14,15].

Since evidence on the association between influenza vaccination and COVID-19 remains conflicting, we aimed to investigate the possible association between influenza vaccination and COVID-19 outcomes in a large cohort of people living in Europe and Israel who participated in the SHARE (Survey of Health, Ageing, and Retirement in Europe) study. 

## 2. Materials and Methods

We used the data from the Survey of Health, Ageing, and Retirement in Europe (SHARE). SHARE is an ongoing longitudinal study of the European population. Complete information about this project can be found in the project’s webpage (http://www.share-project.org/organisation/share-eric.html, accessed 1 April 2022) and has been described in detail elsewhere [16]. Briefly, the SHARE study is a multidisciplinary and cross-national panel database of micro data on health, socio-economic status, and social and family networks of individuals aged 50 or older. SHARE started in 2004 with representative samples of individuals aged 50+. To date, SHARE has conducted nine waves of data collection and has covered all continental EU countries plus Switzerland and Israel. SHARE explores this cross-country setting as a ‘natural laboratory’ across scientific disciplines and over time to turn the challenges of population ageing into opportunities and provide policy makers with reliable information for evidence-based policies.

There is one common generic questionnaire that the country teams translated into the national languages (in some countries more than one language) using an internet-based translation tool. Usually, SHARE data collection is based on computer-assisted personal interviewing (CAPI) because it makes the execution of physical tests possible. The interviewers conduct face-to-face interviews using a laptop on which the CAPI instrument is installed.

### 2.1. Sample and Data

The current analyses used the waves 8 and 9 of the SHARE study, integrated with the information of the SHARE COVID-19 Survey. The SHARE Corona dataset contains data collected via computer-assisted telephone interviews (CATI) in the two rounds of the SHARE Corona Survey (SCS) between June and August 2020 (1st SCS) and one year later between June and August 2021 (2nd SCS) [17]. We included only people aged 50+ years in agreement with other SHARE studies [18,19], since the SHARE study was specifically designed for better understanding of prevention, protection, and treatment information for the population that is aged 50+ years [16].

The final dataset included Israel and all the European countries participating in the SHARE study, i.e., Austria, Germany, Sweden, The Netherlands, Spain, Italy, France, Denmark, Greece, Switzerland, Belgium, Israel, Czech Republic, Poland, Luxembourg, Hungary, Portugal, Slovenia, Estonia, Croatia, Lithuania, Bulgaria, Cyprus, Finland, Latvia, Malta, Romania, Slovakia. 

### 2.2. Exposure: Influenza Vaccination

Information regarding influenza vaccination were collected using the following question: “Did you get the influenza vaccination in the last 12 months?”. The possible answers were “don’t know”, “no”, and “yes”. 

### 2.3. Outcomes: COVID-19

The outcomes of interest for this study were endpoints related to COVID-19 at wave 9, namely: positivity for COVID-19, asking the participant if he/she ever tested positive or not; symptomatic forms in people with positive COVID-19, (i.e., any signs or symptoms typical of COVID-19, such as cough, congestion, shortness of breath, loss of taste or smell, headache, body aches, joint pain, chest or abdominal pain, diarrhea, nausea, or confusion attributed to respondent’s COVID-19 illness); and hospitalization for COVID-19. 

### 2.4. Covariates

For assessing associations between the influenza vaccination and the outcomes of interest, we considered several factors, including country (categorized as mentioned before); age (as continuous); sex; years of formal education (as continuous); smoking status (yes, no); alcohol drinking, assessed by asking the participants how often he/she drank 6 or more drinks during the last 3 months, with answers dichotomized as daily vs. other; current job situation (categorized as retired vs. others); body mass index, calculated using self-reported weight and height. Moreover, information regarding mobility limitation and limitations in activities of daily living (ADL) or instrumental activities of daily living (IADL) were assessed by asking the participants whether they had difficulties in moving or with one or more of six ADLs [20] (dressing, walking across a room, bathing, eating, getting in or out of bed, using the toilet) or with one or more of IADLs [21] (using a map, preparing a meal, shopping for groceries, using the phone, taking medications, doing housework, managing money, using transport, doing personal laundry). Co-morbidities were assessed by asking the participants: “Has a doctor ever told you that you had/do you currently have any of the conditions on this card?”; a show-card with multiple non-mutually exclusive options was presented to the participants. For the aims of this research, we reported the information regarding the following medical conditions: hip fracture, diabetes, high blood pressure, heart problems, lung disease and cancer, other. Similarly, medications were assessed by asking: “Do you currently take drugs, at least once a week, for problems mentioned on this card?”.

Finally, we included self-reported information regarding vaccination against COVID-19, and against pneumococcus during the last six years.

### 2.5. Statistical Analysis

Continuous variables were analyzed in terms of distribution using the Kolmogorov–Smirnov test and using the Levene’s test to test the homoscedasticity of variances. Means and standard deviations (SD) were used to describe continuous measures, while percentages were used for categorical variables. Characteristics of the study participants at baseline (wave 8) were compared according to influenza vaccination status, using Chi-squared or Fisher exact tests for categorical variables, and an independent T-test for continuous variables. To minimize the effect of potential confounders, we used a propensity score matching with one case (received influenza vaccination) and one control without an influenza vaccination, using a tolerance of 0.01 points [22]. Overall, 10,966 participants vaccinated against influenza were matched with 10,966 people who were not vaccinated with this approach. 

The association between influenza vaccination at wave 8 and COVID-19 outcomes at wave 9 was assessed using univariable and multivariable logistic regression analysis and reported as odds ratios (OR) and 95% confidence intervals (95% CI). We reported data adjusted for potential confounders (age, sex, and the list of confounders above) and adjusted using a propensity score matching. In this latter model, we also included country. Finally, to test the robustness of our results, we explored the interactions between influenza vaccination and several factors, including sex, presence of disease or use of medications, and vaccination against COVID-19 and pneumococcus. All statistical tests were two-tailed, and a *p*-value < 0.05 was statistically significant. All analyses were performed using SPSS 26.0 version software.

## 3. Results

Initially, 49,226 participants were included. After removing 287 people younger than 50 years, 143 participants without information regarding influenza vaccination, and 388 without information regarding COVID-19, we included 48,408 participants in this analysis (Figure 1). 

This cohort had a mean age of 67.0 (SD = 9.7; range: 50–104) years, with women constituting 54.1% of the sample. Overall, 18,655 participants received vaccination against influenza in the previous year, corresponding to a prevalence of 38.3% (95%CI: 37.9–38.8%). Comparing the sample by influenza vaccination status, people having received the vaccination were significantly older, more frequently females, and more frequently smokers, alcohol drinkers, and retired than their counterparts (Table 1). Moreover, participants who received an influenza vaccination were significantly leaner, according to their BMI values, and reported a significantly higher limitation in mobility and ADL/IADL as well as a higher prevalence of the comorbidities investigated in the SHARE study, and were more likely to be taking medications. Finally, participants with influenza vaccination reported a significantly higher prevalence of vaccinations against pneumococcus and COVID-19 (Table 1). All these comparisons were statistically significant at a *p*-value < 0.0001. 

During year one of the follow-up, 4028 participants (=8.3% of the baseline population) tested positive for COVID-19. Among them, 3123 reported a symptomatic form and 579 (=1.2% of the baseline population) were hospitalized for COVID-19. Table 2 shows the association between influenza vaccination and COVID-19 outcomes in the SHARE study. Influenza vaccination was associated with a significantly lower risk of all COVID-19 outcomes investigated. After adjusting for 15 potential confounders, including demographic information, comorbidities, and vaccination against COVID-19, influenza vaccination was significantly associated with a lower risk for positivity for COVID-19 (OR = 0.95; 95%CI: 0.94–0.96; *p <* 0.0001), symptomatic forms (OR = 0.87; 95%CI: 0.86–0.88; *p <* 0.0001), and hospitalization for COVID-19 (OR = 0.95; 95%CI: 0.94–0.96; *p <* 0.0001). The results were similar using a propensity score instead of covariates. 

Finally, we tested whether there was effect modification in the association between influenza vaccination and COVID outcomes by several factors, including sex, presence of disease or use of medications, and vaccination against COVID-19 or pneumococcus, and none of these factors emerged as a significant moderator in our analyses.

## 4. Discussion

The present study found that influenza vaccination may have a role in decreasing the onset of overall, symptomatic, and more severe forms of COVID-19. The effect of influenza vaccination on COVID-19 outcomes resulted independently from several potential clinical and demographic factors, including vaccination status and when matching for propensity score.

In the SHARE study, the rate of people vaccinated against influenza was 38.3%, which is somewhat low considering the mean age of the population analyzed, i.e., ≥50 years. These data are, unfortunately, similar with other epidemiological information available across the world, indicating an overall low global rate of vaccination against influenza [23]. At the same time, the recent COVID-19 epidemic has raised the adherence to influenza vaccination in several countries since it was reported that a higher adherence to influenza vaccination could be associated with a lower severity of COVID-19, particularly in the phases in which vaccination against SARS-CoV-2 was not yet available [24,25]. In our study, we found that this hypothesis could be possible, since influenza vaccination may protect against COVID-19. 

We believe that our findings are important because the literature regarding influenza vaccination on COVID-19 outcomes has been limited. Among 3520 healthcare Italian workers, no significant association was found between receipt of flu vaccination in the last five years and identification of COVID-19 using a serology test or any form of COVID-19 [11]. Another Italian study reported that influenza vaccination was associated with a significantly lower risk of overall mortality [12]. However, before our study, no research examined the potential association between influenza vaccination and overall, symptomatic or severe forms of COVID-19. Altogether, our data indicate that an association between flu vaccine and COVID-19 outcome is present, independently from several potential confounders. 

From a pathophysiological point of view, the suggested association between influenza vaccination and a favorable COVID-19 outcome might be due to an immunopathogenic mechanism known as trained immunity. Netea and colleagues showed that several vaccinations, such as flu vaccination, can trigger an adaptive immune response via T-helper cells, which results in memory cellular (macrophages, NK cells) and humoral (antibody-mediated) responses that kill antigen-presenting cells on subsequent exposure to the same antigen [26]. However, other authors demonstrated how innate immune cells’ epigenetic and metabolic reprogramming can lead to a second, non-specific stimulation that might elicit a focused and heightened proinflammatory response [27]. This "heterologous immunity" might explain vaccines’ nonspecific cross-reactivity with diseases that are apparently unrelated. In fact, this immune phenomenon is the basis of action of the bacilli Calmette–Guérin (BCG) vaccination against cancer and malaria, as shown in a recent trial, due to trained immunity [28], and this may also have a potential role in SARS-CoV-2 protection. Finally, another explanation of our findings is derived from recent experience that demonstrated how influenza vaccination influences cytokine production capacity. In fact, as Debisarun and colleagues highlight, influenza vaccinations modulate the responses against SARS-CoV-2, reducing IL-1β and IL-6 production while enhancing IL-1Ra release [29], and the cytokine storm due to dysregulated immune responses is well-known to underlie systemic SARS-CoV-2 damage [30]. However, more studies are, however, necessary to understand any potential role of trained immunity with influenza vaccine and COVID-19 infection. 

The major point of strength of this study is undoubtedly the large cohort included, being representative of Europe and Israel. The results of this study must be, however, interpreted within its limitations. First, all the information regarding exposure, potential confounders, and outcomes were self-reported. In particular, information regarding COVID-19 were asked to the participants and not verified with other tools, such as nasopharyngeal swabs: therefore, an underestimation of the outcomes of interest is possible, particularly for asymptomatic forms. However, these data are similar to those reported in Europe [31]. Future research using confirmed diagnosis of COVID-19 and standardized data regarding influenza vaccination are needed. Second, influenza vaccination was recorded only once (i.e., past 12 months), whilst we have evidence, even if conflicting, that the repetition of this intervention over the years may have positive effects on the protection of some health outcomes, e.g., on dementia incidence [32]. Third, using a propensity-score matching that permits to minimize potential confounding factors, we were able to only select a limited part of the controls, possibly introducing a selection bias. Finally, the data collected refer to a specific time point in the pandemic (first wave).

## 5. Conclusions

Our work, which includes approximately 50,000 European older participants, indicates that influenza vaccination could be beneficial for the prevention of COVID-19, being associated with a lower risk of COVID-19 outcomes. Our findings suggest that longitudinal cohort studies are now needed to investigate the effect of influenza vaccination on COVID-19.

## Figures and Tables

**Figure 1 vaccines-10-00899-f001:**
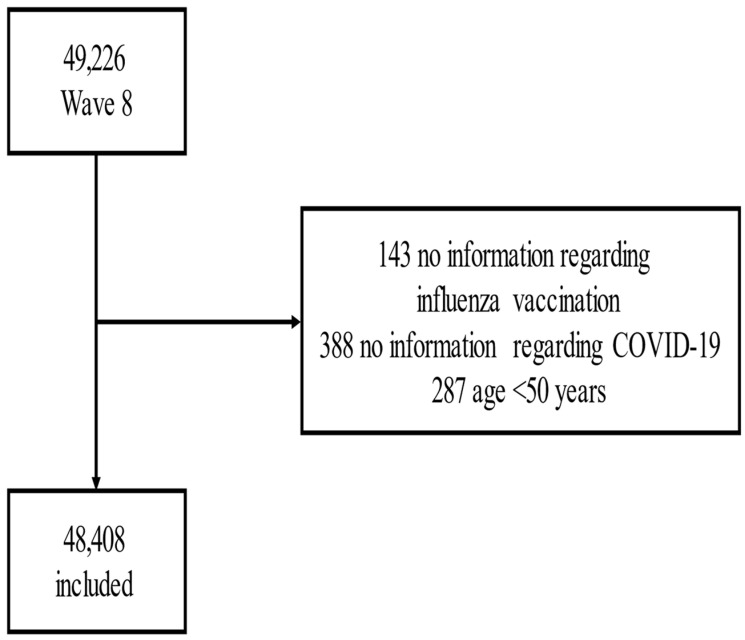
Flow-chart.

**Table 1 vaccines-10-00899-t001:** Descriptive characteristics by influenza vaccination or not.

Variable	Overall Sample	Influenza Vaccination (n = 18,655)	No Influenza Vaccination (n = 29,753)	*p*-Value
**Demographics**				
**Mean age (SD)**	67.0 (9.7)	70.5 (9.7)	64.9 (9.0)	<0.0001
**Females (%)**	54.1	54.2	54.1	<0.0001
**Current smokers (%)**	11.0	8.0	12.9	<0.0001
**Daily alcohol drinking (%)**	1.7	1.9	1.7	<0.0001
**Mean years of education (SD)**	11.4 (4.3)	11.3 (4.6)	11.5 (4.2)	<0.0001
**Retired (%)**	34.7	63.0	44.1	<0.0001
**Medical and functional information**				
**Mean BMI (SD)**	27.0 (4.7)	26.9 (4.6)	27.0 (4.8)	<0.0001
**Mean mobility limitation (SD)**	1.53 (2.28)	1.76 (2.38)	1.37 (2.20)	<0.0001
**Mean limitation in ADL (SD)**	0.21 (0.81)	0.27 (0.92)	0.18 (0.73)	<0.0001
**Mean limitations in IADL (SD)**	0.40 (1.32)	0.51 (1.51)	0.34 (1.18)	<0.0001
**Hip fracture (%)**	0.3	0.3	0.3	<0.0001
**Diabetes (%)**	1.6	1.9	1.5	<0.0001
**High blood pressure (%)**	3.9	4.6	3.6	<0.0001
**Heart problems (%)**	2.1	2.7	1.8	<0.0001
**Lung disease (%)**	1.1	1.5	0.8	<0.0001
**Cancer (%)**	1.3	2.1	0.8	<0.0001
**Regularly takes prescription drugs (%)**	69.5	82.7	63.4	<0.0001
**Vaccination history**				
**Vaccination against COVID-19 (%)**	82.5	95.5	74.4	<0.0001
**Vaccination against pneumococcus (%)**	11.5	23.8	3.8	<0.0001

**Table 2 vaccines-10-00899-t002:** Association between influenza vaccination and COVID-19 outcomes.

Influenza Vaccination	Percentage of Events	Fully-Adjusted ^1^ OR, 95% CI	Propensity Score ^2^ Adjusted OR, 95% CI
**Positivity for COVID-19**			
No	4.8	1 [reference]	1 [reference]
Yes	6.2	0.95 (0.94–0.96) *p* ≤ 0.0001	0.86 (0.85–0.87) *p* ≤ 0.0001
**Symptomatic COVID-19**			
No	8.1	1 [reference]	1 [reference]
Yes	5.8	0.87 (0.86–0.88) *p* ≤ 0.0001	0.96 (0.96–0.97) *p* ≤ 0.0001
**Hospitalization COVID-19**			
No	1.1	1 [reference]	1 [reference]
Yes	0.9	0.95 (0.94–0.96) *p* ≤ 0.0001	0.78 (0.77–0.79) *p* ≤ 0.0001

^1^ Adjusted for country, mean age, sex, smoking status, alcohol drinking, mean years of education, job status, mean body mass index, mobility limitation, limitations in basic and instrumental activities of daily living, presence of any disease, use of medication, and vaccination against pneumococcus and against COVID-19. ^2^ The propensity score was included in the model as a continuous variable. Country was forced in this model.

## Data Availability

The data presented in this study are available on request from the corresponding author.

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
