# Peer review of "Influenza Vaccination and COVID-19 Outcomes in People Older than 50 Years: Data from the Observational Longitudinal SHARE Study"

_vaccines, 2022, doi:10.3390/vaccines10060899_

Round 1

Reviewer 1 Report

This article presents interesting findings but it suffers from various limitations that must be addressed before it is accepted for publication.

Some comments: 

  • I suggest specifying (in the title) that this work was conducted among adults (50+) and it represents an observational study.
  • The introduction is nicely written but does not mention important secondary studies that have already investigated the association between influenza vaccination and COVID-19 outcomes (1-3).They should be considered to give the reader a proper overview of the current evidence.
  • I suggest specifying what "wave 8" and "wave 9" are referred to (line 95). 
  • The way the sampling and data collection is described is insufficient, I suggest explaining how the adults were selected and - above all - data collected. It is important as the whole study relies on self-reported data, so the process should be extensively explained. 
  • As you correctly mentioned, all the information is self-reported. This limits the internal validity of the study. It is important that you also mentioned it in the abstract, it timely informs the reader about it. However, I suggest expanding the limitation section and better discussing potential biases you may have incorporated in your analysis, suggesting how further studies should be conducted to avoid the same biases
  • (lines 267-270): whether repeated influenza vaccination over the years may have a higher clinical effect is still debated, as there is conflicting evidence. I suggest just reporting that you do not have information about previous history of influenza vaccination or influenza infections, and this is a limit as there are evidence - despite conflicting - that it can have effects on protection. 
  • Table 1 is not that clear. As an example, Please report (table 1 or supplementary table) the actual figures for each variable. 

  1. Del Riccio M, Lorini C, Bonaccorsi G, Paget J, Caini S. The Association between Influenza Vaccination and the Risk of SARS-CoV-2 Infection, Severe Illness, and Death: A Systematic Review of the Literature. Int J Environ Res Public Health. 2020;17(21):7870. Published 2020 Oct 27. doi:10.3390/ijerph17217870
  2. Wang R, Liu M, Liu J. The Association between Influenza Vaccination and COVID-19 and Its Outcomes: A Systematic Review and Meta-Analysis of Observational Studies. Vaccines (Basel). 2021;9(5):529. Published 2021 May 20. doi:10.3390/vaccines9050529
  3. Su W, Wang H, Sun C, et al. The Association Between Previous Influenza Vaccination and COVID-19 Infection Risk and Severity: A Systematic Review and Meta-analysis [published online ahead of print, 2022 Mar 15]. Am J Prev Med. 2022;S0749-3797(22)00131-3.

Author Response

This article presents interesting findings but it suffers from various limitations that must be addressed before it is accepted for publication.

R: We sincerely thank the Reviewer for his /her comments that further improved our work.

Some comments: 

  • I suggest specifying (in the title) that this work was conducted among adults (50+) and it represents an observational study.

R: Added.

  • The introduction is nicely written but does not mention important secondary studies that have already investigated the association between influenza vaccination and COVID-19 outcomes (1-3).They should be considered to give the reader a proper overview of the current evidence.

R: We really thank the Reviewer for this comment. We have added in the Introduction section a sentence regarding the most important findings of these relevant works:

“More recently, some systematic reviews reported that influenza vaccination was not associated with any increased risk of COVID-19 infection [13] or with a reduced incidence of COVID-19, but not to a decreased risk in more severe forms.[14,15]

  • I suggest specifying what "wave 8" and "wave 9" are referred to (line 95). 

R: Done.

  • The way the sampling and data collection is described is insufficient, I suggest explaining how the adults were selected and - above all - data collected. It is important as the whole study relies on self-reported data, so the process should be extensively explained. 

R: We sincerely thank the Reviewer for this comment. We have now added this paragraph, as suggested:

“Briefly, the SHARE study is a multidisciplinary and cross-national panel database of micro data on health, socio-economic status and social and family networks of individuals aged 50 or older. SHARE started in 2004 with representative samples of individuals aged 50+. To date, SHARE conducted nine waves of data collection and covers all continental EU countries plus Switzerland and Israel. SHARE explores this cross-country setting as a ‘natural laboratory’ across scientific disciplines and over time in order to turn the challenges of population ageing into opportunities and provide policy makers with reliable information for evidence based policies.

There is one common generic questionnaire that the country teams translate into the national languages (in some countries more than one language) using an internet based translation tool. Usually, SHARE data collection is based on computer-assisted personal interviewing (CAPI) because it makes the execution of physical tests possible. The interviewers conducted face-to-face interviews using a laptop on which the CAPI instrument is installed.”

  • As you correctly mentioned, all the information is self-reported. This limits the internal validity of the study. It is important that you also mentioned it in the abstract, it timely informs the reader about it. However, I suggest expanding the limitation section and better discussing potential biases you may have incorporated in your analysis, suggesting how further studies should be conducted to avoid the same biases

R: As suggested, we added some more information in the Abstract and Limitations section regarding the potential bias of self-reported information.

Abstract

“Positivity for COVID-19, symptomatology and hospitalization were also ascertained using self-reported information.”

Limitations section:

“Future research using confirmed diagnosis of COVID-19 and standardized data regarding influenza vaccination are needed.”

  • (lines 267-270): whether repeated influenza vaccination over the years may have a higher clinical effect is still debated, as there is conflicting evidence. I suggest just reporting that you do not have information about previous history of influenza vaccination or influenza infections, and this is a limit as there are evidence - despite conflicting - that it can have effects on protection. 

R: Thank you for the question. We have added the following limitation:

“Second, influenza vaccination was recorded only once (i.e., past 12 months), whilst we have evidence, even if conflicting, that the repetition of this intervention during the years may positive have effects on the protection of some health outcomes, e.g., on dementia incidence.”

  • Table 1 is not that clear. As an example, Please report (table 1 or supplementary table) the actual figures for each variable. 

R: Sorry for the inconvenience. We have now added a column in Table 1 regarding the data of the whole sample included.

  1. Del Riccio M, Lorini C, Bonaccorsi G, Paget J, Caini S. The Association between Influenza Vaccination and the Risk of SARS-CoV-2 Infection, Severe Illness, and Death: A Systematic Review of the Literature. Int J Environ Res Public Health. 2020;17(21):7870. Published 2020 Oct 27. doi:10.3390/ijerph17217870
  2. Wang R, Liu M, Liu J. The Association between Influenza Vaccination and COVID-19 and Its Outcomes: A Systematic Review and Meta-Analysis of Observational Studies. Vaccines (Basel). 2021;9(5):529. Published 2021 May 20. doi:10.3390/vaccines9050529
  3. Su W, Wang H, Sun C, et al. The Association Between Previous Influenza Vaccination and COVID-19 Infection Risk and Severity: A Systematic Review and Meta-analysis [published online ahead of print, 2022 Mar 15]. Am J Prev Med. 2022;S0749-3797(22)00131-3.

Reviewer 2 Report

The manuscript describes a secondary data analysis on the association between influenza vaccination and COVID-19 outcomes. The sample size is sufficiently large with appropriate statistical analyses. My comments are listed below:

1.The authors also remind readers about their study limitations (mainly, all data were self-reported).

2. The authors also reported that they used propensity score matching to minimize potential confounding factors. However, there were 10966 cases who were vaccinated, the total sample size was less than 50,000. As a result, the selection of controls is limited.

3. The overall influenza vaccination rate was 38.3%. Is that higher or lower than those of previous years among the same populations where survey participants came from. 

Author Response

The manuscript describes a secondary data analysis on the association between influenza vaccination and COVID-19 outcomes. The sample size is sufficiently large with appropriate statistical analyses. My comments are listed below:

1.The authors also remind readers about their study limitations (mainly, all data were self-reported).

R: Thank you for this suggestion. The following information can be found in the limitation section.

“First, all the information regarding exposure, potential confounders, and outcomes were self-reported. In particular, information regarding COVID-19 were asked to the participants and not verified with other tools, such as nasopharyngeal swabs: therefore, an underestimation of the outcomes of interest is possible, particularly for asymptomatic forms.”

  1. The authors also reported that they used propensity score matching to minimize potential confounding factors. However, there were 10966 cases who were vaccinated, the total sample size was less than 50,000. As a result, the selection of controls is limited.

R: We fully agree regarding this important aspect. We have added a limitation in the Discussion section:

“Third, we were able to only select a limited number of the controls for propensity-score matching, and it is possible that this introduced a selection bias.”

  1. The overall influenza vaccination rate was 38.3%. Is that higher or lower than those of previous years among the same populations where survey participants came from. 

R: Good point. Unfortunately, as stated, we don’t have information regarding influenza vaccination in the previous years.

Round 2

Reviewer 1 Report

Thank you for addressing all the comments.